# Unveiling the Microbiome Diversity in *Telenomus* (Hymenoptera: Scelionidae) Parasitoid Wasps

**DOI:** 10.3390/insects15070468

**Published:** 2024-06-23

**Authors:** Mayra A. Gómez-Govea, Kenzy I. Peña-Carillo, Gabriel Ruiz-Ayma, Antonio Guzmán-Velasco, Adriana E. Flores, María de Lourdes Ramírez-Ahuja, Iram Pablo Rodríguez-Sánchez

**Affiliations:** 1Laboratorio de Fisiología Molecular y Estructural, Facultad de Ciencias Biológicas, Universidad Autónoma de Nuevo León, San Nicolás de los Garza 64460, Mexico; mayragee@gmail.com; 2Campo Experimental General Terán, Instituto Nacional de Investigaciones Forestales Agrícolas y Pecuarias, Km 31 Carretera Montemorelos-China, General Terán 67400, Mexico; kenzy.p@gmail.com; 3Laboratorio de Conservación de Vida Silvestre y Desarrollo Sustentable, Facultad de Ciencias Biológicas, Universidad Autónoma de Nuevo León, San Nicolás de los Garza 64460, Mexico; gabriel.ruizym@uanl.edu.mx (G.R.-A.); antonio.guzman@uanl.mx (A.G.-V.); 4Laboratorio de Entomología Médica, Departamento de Zoología de Invertebrados, Facultad de Ciencias Biológicas, Universidad Autónoma de Nuevo León, San Nicolás de los Garza 66455, Mexico; adriana.floressr@uanl.edu.mx

**Keywords:** wasp parasitoids, microbiome, *Telenomus* species

## Abstract

**Simple Summary:**

This study was undertaken to determine the gut microbiota of six species of the *Telenomus* genus (*T. alecto*, *T. sulculus*, *T. fariai*, *T. remus*, *T. podisi*, and *T. lobatus*). Wasp parasitoids were collected from their hosts in different locations in Mexico. DNA was extracted from gut collection, and sequencing of bacterial 16S rRNA was carried out in Illumina^®^ MiSeq™. Among the six species of wasps, results showed that the most abundant phylum were Proteobacteria (82.3%), Actinobacteria (8.1%), and Firmicutes (7.8%). The most important genera were *Delftia* and *Enterobacter*. Seventeen bacteria species were found to be shared among the six species of wasps.

**Abstract:**

Bacterial symbionts in insects constitute a key factor for the survival of the host due to the benefits they provide. Parasitoid wasps are closely associated with viruses, bacteria, and fungi. However, the primary symbionts and their functions are not yet known. This study was undertaken to determine the gut microbiota of six species of the *Telenomus* genus: *T. alecto* (Crawford), *T. sulculus* Johnson, *T. fariai* Costa Lima, *T. remus* Nixon, *T. podisi* Ashmead, and *T. lobatus* Johnson & Bin. Wasp parasitoids were collected from their hosts in different locations in Mexico. DNA was extracted from gut collection, and sequencing of bacterial 16S rRNA was carried out in Illumina^®^ MiSeq™. Among the six species of wasps, results showed that the most abundant phylum were Proteobacteria (82.3%), Actinobacteria (8.1%), and Firmicutes (7.8%). The most important genera were *Delftia* and *Enterobacter*. Seventeen bacteria species were found to be shared among the six species of wasps. The associate microbiota will help to understand the physiology of *Telenomus* to promote the use of these wasp parasitoids in the management of insect pests and as potential biomarkers to target new strategies to control pests.

## 1. Introduction

Wasps of the family Scelionidae (Hymoptera: Telenominae) are parasitoids of other insects and arachnid eggs. The family is divided into three subfamilies: Scelioninae, Teleasinae, and Telenominae [1]. The subfamily Telenominae is important because it contains the species used for the biological control of insect pests of the orders Hemiptera and Lepidoptera, noxious for agriculture pests, forestry, and medical importance [2]. In particular, species of *Telenomus* parasitize insects of the orders Hemiptera, Lepidoptera, and Neuroptera. For example, the species *T. remus* has been reported to parasitize *Spodoptera frugiperda* (Smith) eggs [3], while the species *T. alecto* has been associated with the parasitization of *Diatrea magnifactella* (Lepidoptera: Crambidae) eggs [4].

In recent years, gut microbiota has been the focus of studies for a wide variety of organisms, including insects, revealing that it plays a key role in host nutrition, detoxification of toxic compounds, and reproduction [5,6,7]. Several studies of wasps have been focused on the *Nasonia* genera, which have determined the fundamental role of the intestinal microbiome for survival [8,9,10]. 

Intestinal colonization of insects by bacteria depends on several factors, including taxonomy, evolution, ecology, diet, and host life stage [11]. The core microbiome is an essential group of host-associated microorganisms notable for their biological function [12]. Many studies have shown the presence of bacteria essential for insects; for example, the bacteria *Buchnera* supplies essential amino acids and vitamins for aphids [13], and *Blochmannia* has been found as an obligate intracellular endosymbiont in ants of the genus *Camponotus* [14]. Currently, using the intracellular symbiont *Wolbachia pipientis* as a manipulator of mosquito reproduction has raised interest in its application in the biological control of its populations [15]. In hymenopterans, the presence of endosymbiotic fitness-impairing bacteria, such as *Rickettsia*, *Arsenophonus*, and *Wolbachia*, has been studied [16,17]. Bacteria in bees, such as *Apis mellifera* L., have been associated with health and nutrition [18,19,20]. In recent studies, the microbiome of the carpenter bee (*Ceratina calcarata*) was identified with thirteen central bacteria important for their physiology: *Rosenbergiella*, *Pseudomonas*, *Gilliamella*, *Lactobacillus*, *Caulobacter*, *Snodgrassella*, *Acinetobacter*, *Corynebacterium*, *Sphingomonas*, *Commensalibacter*, *Methylobacterium*, *Massilia*, and *Stenotrophomonas* [21]. In addition, bumblebees are known to harbor a simple and distinct gut microbiome comprised of core bacterial species within the genus *Snodgrassella*, *Gilliamella*, *Lactobacillus*, *Bombiscardovia*, *Schmidhempelia*, and *Bifidobacterium*, which defend them against pathogens [19,22]. 

Little information exists about the key bacteria of wasps of the genus *Telenomus* and their physiological importance. The study of key bacteria can be essential to understanding the basic biology of the organism and its interactions with the host [23]. The objective of this study was to determine the microbiome of six species of *Telenomus: T. alecto*, *T. sulculus*, *T. fariai*, *T. remus*, *T. podisi*, and *T. lobatus,* to understand physiological interactions between the organisms and their associated microbiome in the host.

## 2. Materials and Methods

The specimens were obtained from their hosts in different regions across Mexico (Figure 1). *Telenomus alecto* was obtained after it emerged from the eggs of *Diatrea magnifactella*, *T. remus* from *Spodoptera frugiperda*, *T. fariai* from *Triatoma dimidiata* (Latreille) (Hemiptera, Reduviidae), *T. podisi* from *Arvelius albopunctatus* (DeGeer) (Hemiptera: Pentatomidae), *T. sulculus* from *Zelus renardii* Kolenati (Hemiptera: Reduviidae), and *T. lobatus* from Chrysopidae. Eggs of all hosts were taken to the laboratory (HR 70%, T 25 ± 2 °C) and placed into Petri dishes. Observations were made until parasitoids emerged. The sex ratio of the parasitoids was recorded.

### 2.1. Gut Collection 

Guts from eight-day-old adults were used for tissue isolation. Before dissection, adult insects were surface sterilized with 96% ethanol for three minutes and rinsed with sterile deionized water thrice. Sixty female adults of each species were dissected on a plate that contained 2 mL of sterile phosphate-buffered solution (10 mmol/L, pH 7.4, Ambion, Thermo Fisher Scientific, Waltham, MS, USA) using a pair of sterilized needles (flame-sterilized and cooled using 96% ethanol) under a stereomicroscope (Leica MZ16, 1.6×). Guts were deposited in plastic Eppendorf^®^ 1.5 mL tubes with DNA shield and stored at −70 °C for DNA extraction and sequencing. 

### 2.2. Extraction of DNA

The samples used in this study were analyzed using the ZymoBIOMICS^®^ service performed by Zymo Research (Irvine, CA, USA). DNA was extracted from samples with the ZymoBIOMICS^®^-96 MagBead DNA Kit (Zymo Research, Irvine, CA, USA) and processed with the ZymoBIOMICS™ Service–Targeted Metagenomic Sequencing (Zymo Research, Irvine, CA, USA). Four replicates were made for each *Telenomus* species. Bacterial 16S rRNA gene-targeted sequencing was performed using the Quick-16S™ NGS Library Prep Kit (Zymo Research, Irvine, CA, USA). Bacterial 16S primers amplified the V1–V2 or V3–V4 regions of the 16S rRNA gene. The sequencing library was prepared according to Gómez-Govea et al. [24], and the final library was sequenced on Illumina^®^ MiSeq™, San Diego, CA, USA. 

### 2.3. Bioinformatics Analysis

Raw reads were quality-filtered to remove low-quality data and chimeric sequences using the Dada2 pipeline [25]. Reads were clustered into operational taxonomic units (OTUs) with representative sequences and read counts (abundances) into operational taxonomic units at 97% (species-level) sequence identity to compare OTU abundance between wasp species. If an OTU contained at most five reads, it was omitted from downstream analyses. The resulting data were analyzed using the Quantitative Insights Into Microbial Ecology (Qiime v.1.9.1) pipeline for taxonomy assignment (Greengenes database as reference (http://greengenes.lbl accessed on 12 June 2023)), alpha-diversity (Shannon diversity), and beta-diversity (Chao1) [26]. To compare the microbial community differences among different species of wasps, a principal coordinates analysis (PCoA) on Bray–Curtis distances using Qiime v.1.9 was performed. Other analyses, such as heat maps and abundance plots, were performed with internal scripts.

## 3. Results

The bacterial community of six species of parasitoid wasps of *Telenomus* was determined. The results showed twelve phyla; the most representative phyla were Proteobacteria (82.3%), Actinobacteria (8.1%), and Firmicutes (7.8%) (Figure 2A). Twenty-one classes were identified, highlighting the classes Gammaproteobacteria (45.7%), Betaproteobacteria (32.8%), Actinobacteria (8.1%), Bacilli (6.2%), and Alphaproteobacteria (3.7%) (Figure 2B). The orders Enterobacteriales (35.2%) and Burkholderiales (31.9%) were present at high levels in all the wasps. In comparison, Corynebacteriales (10.3%), Pseudomonadales (8.0%), Propionibacteriales (4%), Bacillales (3.7%), Lactobacillales (2.6%), Rickettsiales (2.1%), Pasteurellales (1.7%), and Clostridiales (1.4%) were other orders that were found (Figure 3). This study mainly highlighted two families: Comamonadaceae (31.3%) and Enterobacteriaceae (35.2%). At the genus level, it was dominated by *Delftia* (31.1%), *Enterobacter* (33%), *Moraxella* (4%), Pseudomonas (2.0%), and *Acinetobacter* (1.8%).

In addition, the results showed other genera with percentages above 1% that were not present in all wasp species studied. *Staphylococcus* (2.8%) and *Corynebacterium* (1.6%) were found only in four wasp species (*T. alecto*, *T. sulculus*, *T. fariai*, and *T. lobatus*); *Cutibacterium* (4.0%) in three species (*T. alecto*, *T. fariai*, and *T. lobatus*); *Wolbachia* (1.6%) in two species (*T. sulculus* and *T. podisi*); *Actinobacillus* (1.5%) in four species (*T. alecto*, *T. sulculus*, *T. remus*, and *T. podisi*); *Salmonella* (1.4%) in five species (*T. sulculus*, *T. fariai*, *T. remus*, *T. podisi*, and *T. lobatus*); and *Streptococcus* (1.3%) in four species (*T. alecto*, *T. fariai*, *T. podisi,* and *T. lobatus*). Three hundred bacteria species have been detected in this study, among which 17 bacteria species were found in all *Telenomus* species (Figure 4).

Sixty-four percent of the bacteria genus corresponded to *Enterobacter* (*Enterobacter asburiae*, *Enterobacter ludwigii*, (Figure 5); and the complex *Enterobacter cloacae-hormaechei-ludwigii*) and *Delftia* (*Delftia lacustris-tsuruhatensis* complex) (Figure 6). 

In addition, other species were detected, such as *Cutibacterium acnes*, *Wolbachia pipientis*, *Salmonella enterica*, *Staphylococcus epidermidis-hominis*, *Actinobacillus capsulatus*, *Moraxella cuniculi,* and *Pseudomonas otitidis* (Figure 7).

The analysis revealed a statistically significant difference (*p* < 0.05) in diversity, measured by the Shannon index, between parasitoid species *T. alecto* (95%CI = 2.732–4.800), *T. sulculus* (95%CI = 2.914–4.879), *T. fariai* (95%CI = 2.642–4.991), *T. lobatus* (95%CI = 2.669–4.739), *T. remus* (95%CI = 2.006–3.488), and *T. podisi* (95%CI = 2.552–4.060). The PCoA plot with Bray–Curtis distance comparison showed a difference between the bacterial communities of the wasp (F = 6.38, *p* = 0.009) grouping into two main clades, where *T. alecto* and *T. lobatus* presented similarities in their bacterial profiles. In the same way, *T. remus* and *T. podisi* presented similarities between these species. The difference in distances increased with the species *T. fariai* and *T. sulculus*, which are located outside the similarity clades (Figure 8).

## 4. Discussion

The contribution of intestinal microorganisms to the function of insects is very relevant in various aspects, for example, in medicine, agriculture, and ecology [5]. Insect parasitoids harbor microbial communities, including viruses, bacteria, and fungi [7,27]. To date, there is limited evidence about the gut microbiota of parasitoid wasps, especially those of the genus *Telenomus.* Here, we examined the gut microbiota of six species: *T. alecto*, *T. sulculus*, *T. fariai*, *T. remus*, *T. podisi*, and *T. lobatus,* to find key bacteria.

Our results showed the phyla Proteobacteria (82.3%), Actinobacteria (8.1%), and Firmicutes (7.8%) in all *Telenomus* species. Studies of bacterial diversity in *Nasonia* have shown the presence of Proteobacteria (74.4%), Actinomyces (15.7%), and Firmicutes (9.5%) [10]. Ramírez-Ahuja et al. [28] reported the microbiomes of *Telenomus tridentatus* (Platygastroidea: Scelionidae) (Proteobacteria, Actinobacteria, and Firmicutes), which were similar to the results obtained at the phylum level.

It is known that the gut microbiome of insects consists of specialized symbionts, which are considered core residents [29]. These microbial communities vary widely among insects; factors such as the environment, diet, stage of development, and host phylogeny are determinants of their establishment [7]. In this study, 72% of the bacterial community found in *Telenomus* species was dominated by the genus *Delftia* (31.1%), *Enterobacter* (33%), *Moraxella* (4%), *Pseudomonas* (2.0%), and *Acinetobacter* (1.8%). *Delftia* is a diverse genus of Betaproteobacteria widely distributed in the environment with ecological versatility; many of its strains have agricultural and industrial relevance, including plant growth promotion, bioremediation of hydrocarbon-contaminated soils, and heavy metal immobilization [30]. Particularly, *Delftia* species have been related to the ability to degrade organophosphates [31], and some strains are associated with degradation routes for di-n-butylphthalate (DBP, industrial pollutant) [32]. In the same way, these bacteria can produce bacteriocins against some bacterial and fungal pathogens [33,34,35,36]. This genus has been found in the coffee insect pest *Hypothenemus hampei* Ferrari (Coleoptera: Curculionidae) [37] and was isolated from the gut of chlorpyrifos ethyl-resistant insect larvae [38]. Also, it has been associated with chironomid larvae [39] and with the *Homalodisca vitripennis* (Hemiptera: Cicadellidae) microbiome [32,40], *Colaphellus bowringi* (Coleoptera: Chrysomelidae), *Hypothenemus hampei*, *Periplaneta americana* L. [34], and *Aedes aegypti* (L.) mosquitoes [24]. Due to the functions reported in other insects, *Delftia* could be a key bacterium for future studies to determine the importance of physiological mechanisms in *Telenomus* species.

Another highly represented genus in the study was *Enterobacter*. The species of this genus have been recognized as inhabitants in the guts of several insect species [41,42,43]. Many Enterobacter strains are not simply insect commensals but confer beneficial traits to their hosts that primarily fall into two categories: provision and degradation of nutrients and protection from pathogens [44]. *Enterobacter* species also serve as dietary supplements (probiotics) for larvae diets. For example, the Mediterranean fruit fly *Ceratitis capitata* (Diptera: Tephritidae) has been used as a probiotic strain of *Enterobacter* sp. and improved pupal and adult production [43,45].

Microbiome bacterial densities are important to avoid an imbalance in the microbiota [46]. Primary symbionts are essential for specific insect functions, such as synthesizing toxins, modulating the insect immune system, and protecting against natural enemies (viruses, bacteria, and parasitoids) [6,47]. In the case of the *Telenomus* species, since there is no information about their primary or secondary symbionts, we encourage conducting further studies to demonstrate the function of *Delftia* and *Enterobacter* in these wasps.

In this study, we found some species of bacteria, such as *Wolbachia pipientis* in *T. sulculus* and *T. podisi.* In *Telenomus nawai*, this bacterium has been shown to induce parthenogenesis, cause alterations from crossover incompatibility (cytoplasmic incompatibility), kill male offspring (male slaughter), convert genetic males into functional females (feminization), and induce full parthenogenetic reproduction, or thelithoky (induction of parthenogenesis) [48,49]. We observed that during the emergence of *T. sulculus*, 95% of the adults that emerged were females and the remaining were males, leading us to hypothesize that this could be due to the presence of *Wolbachia*.

Another bacterium found in our study was *Pseudomonas*. This genus has been related to other insects, such as the bark beetle (Coleoptera: Curculionidae: Scolytinae), where it contributes to its defense through nutrient competition and the inhibition of entomopathogenic fungi, as well as the provision of nutrients [50]. The genus *Pseudomonas* has been reported to be necessary at low densities. Still, if there is an imbalance caused by the environment or suppression of the microbiota, these bacteria could colonize, increase their density, and affect host fitness [24]. Among the species found in the microbial communities of *Telenomus*, some bacteria can become pathogens in domestic and wild animals [51,52,53].

The microbiome could develop an intimate relationship with the insect host in wasp parasitoids. There is increasing evidence that host–parasite interactions are influenced by microbes associated with the host [27]. Many microorganisms are transient and acquired from the environment or from interaction with the host. This has been demonstrated in insects such as Heteroptera, where extracellular bacteria are left by females on the surface of the egg in secretions or feces, which are then acquired by hatching nymphs [54]. Wasp parasitoids colonize their host, often ingested by bacteria on the outside. These bacteria can either colonize or transit through the gut [55,56]. In our results, we have six species that parasitize different hosts. However, the most abundant bacteria were distributed in all species; there is some difference in less abundant bacteria, which may be due to the host that they parasitize. The relationship between these bacteria and their host remains to be investigated to test if they have been evolutionarily conserved and if there is a phylosymbiosis.

## 5. Conclusions

Parasitoid insects are important components of terrestrial ecosystems in terms of biodiversity and ecological impact [27]. As the utilization of insecticides has created environmental pollution and health problems, using biological control agents such as wasp parasitoids is a promising alternative [57,58]. The technology for mass rearing of parasitoids is well-developed, relatively easy, and inexpensive. However, the effectiveness of releases is variable and depends on many factors, including the microbiota [59]. In this study, the microbiota of six species of the *Telenomus* genus was determined. Our findings demonstrate two main bacteria genera in the microbial community (*Delftia* and *Enterobacter*). Determining their functionality within this genus should be the subject of further studies. The resident microbiota of insects has great potential as a biomarker of insect characteristics and a target for new strategies to control pests and manipulate their characteristics.

## Figures and Tables

**Figure 1 insects-15-00468-f001:**
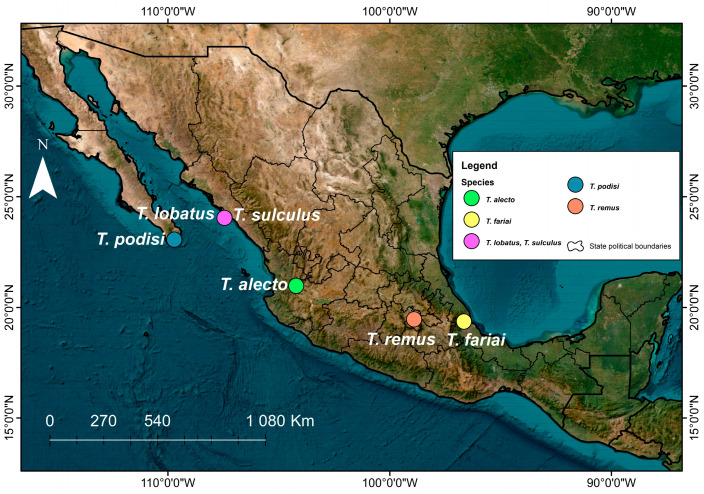
Regions where *Telenomus* species were collected.

**Figure 2 insects-15-00468-f002:**
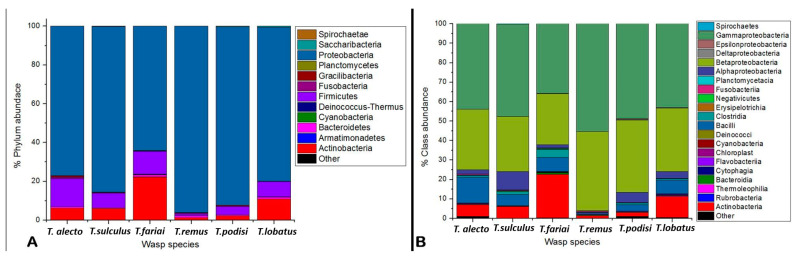
Taxonomic abundance at phylum (**A**), and at class (**B**) levels in *Telenomus* species.

**Figure 3 insects-15-00468-f003:**
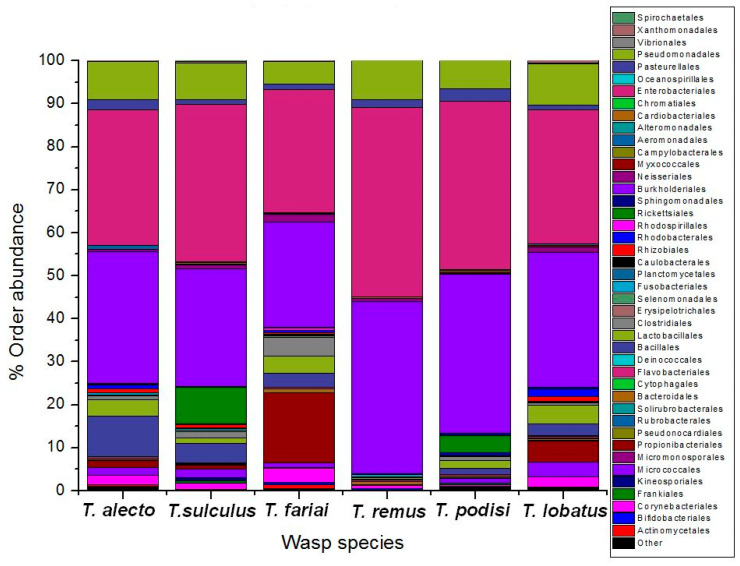
Taxonomic abundance at order level in *Telenomus* species.

**Figure 4 insects-15-00468-f004:**
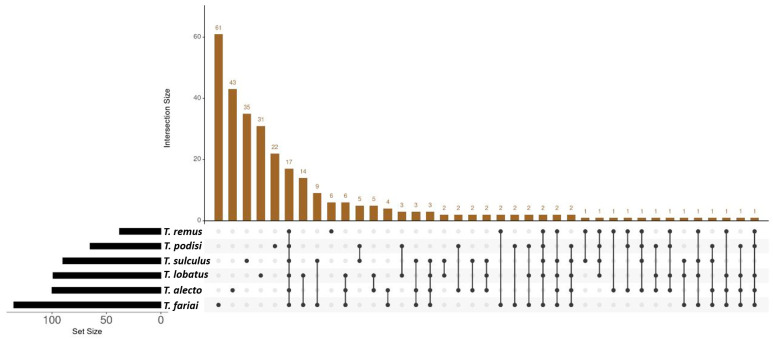
Interaction of bacterial species between species of wasps. The black dots mean the bacteria that are sharing the *Telenomus* species, for example *T. sulculus* and *T. fariai* shared nine species.

**Figure 5 insects-15-00468-f005:**
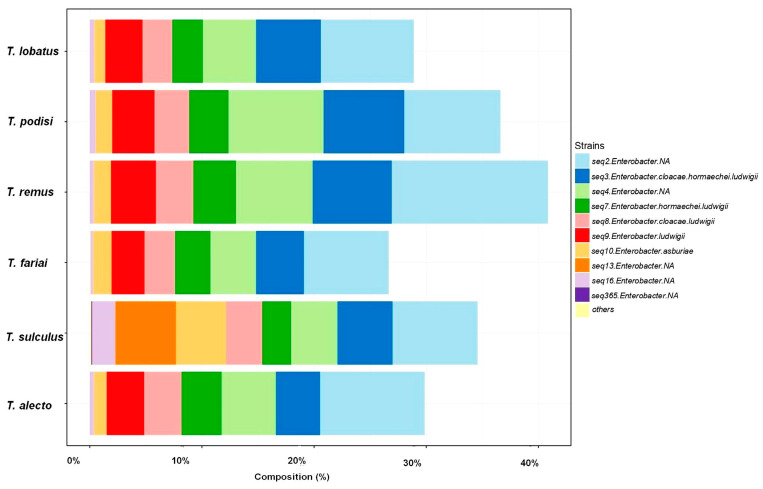
Composition of species of the most representative *Enterobacter* species.

**Figure 6 insects-15-00468-f006:**
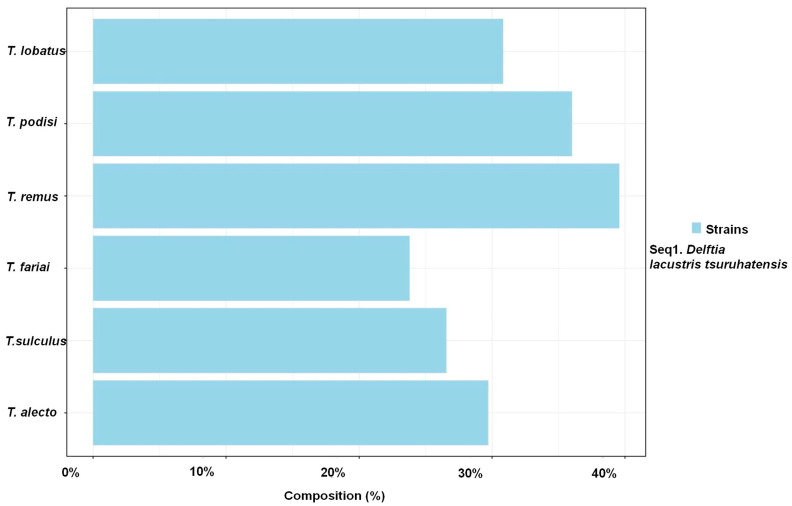
Composition of species of the most representative *Delftia* species.

**Figure 7 insects-15-00468-f007:**
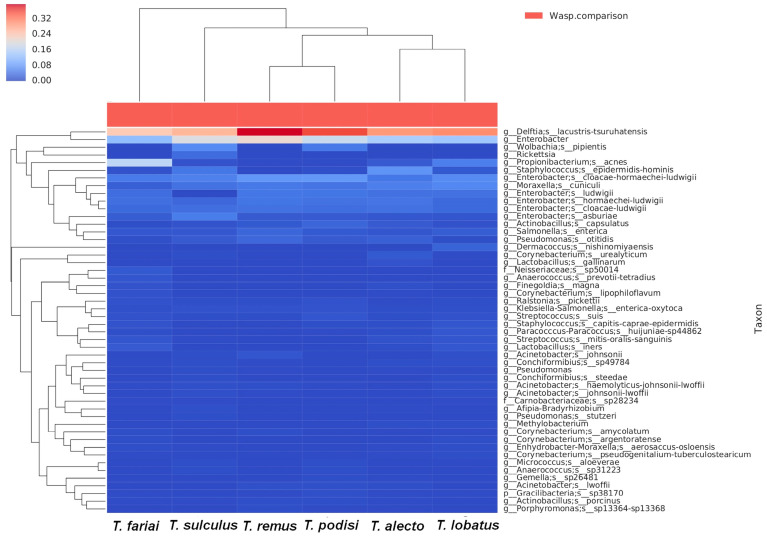
Heat map of species profiles with clustering of top fifty most abundant *Telenomus* species identified.

**Figure 8 insects-15-00468-f008:**
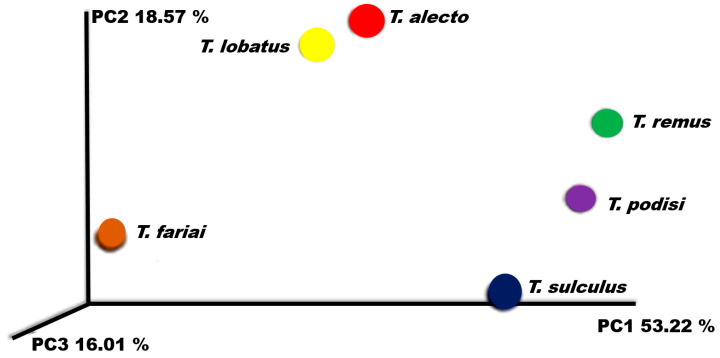
PCoA plot using Bray–Curtis distance showing the distribution of bacterial community composition in *Telenomus* species.

## Data Availability

The datasets presented in this study can be found in online repositories. The names of the repository/repositories and accession number(s) can be found below: National Center for Biotechnology Information (NCBI) BioProject database under accession numbers OR626102-OR626593 entitled “Prokaryotic 16S rRNA/*Telenomus* microbiome”.

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
