# Peer review of "Unveiling the Microbiome Diversity in Telenomus (Hymenoptera: Scelionidae) Parasitoid Wasps"

_insects, 2024, doi:10.3390/insects15070468_

Round 1
Reviewer 1 Report
Comments and Suggestions for Authors
This manuscript describes the elucidation of microbiome biodiversity in the guts of six Telenomus parasitoids collected from Mexico, achieved through sequencing the bacterial 16S rRNA using Illumina high-throughput sequencing technology. The experimental design is reasonable, and the writing is generally well done. However, the data presented are not substantial enough for the manuscript to be published as a full article. I suggest that the manuscript be accepted as a communication after addressing the following issues:
1. It is crucial to collect samples in a sterilized environment to ensure that the results are not contaminated by environmental bacteria. Therefore, the authors should specify that the sample dissection was conducted under sterilized conditions.
2. Biological replicates are essential for drawing confident conclusions. The manuscript lacks information on the number of biological replicates performed for the Illumina sequencing. Detailed methods, including the number of replicates, must be provided in Section 2.2.
3. The Latin names of the parasitoid species displayed in the figures should be italicized.
4. The figures are not of high resolution and need to be improved for clarity.
5. The format of some references must be reviewed and corrected to comply with the journal's requirements.
Reviewer 2 Report
Comments and Suggestions for Authors
A brief summary
Telenomus is a parasitoid wasp in the family of Scelionidae. The authors of this manuscript conducted a study to uncover the gut microbiota of several species of Telenomus. From the abstract, introduction, materials and methods to results, discussions and conclusions, they were presented clearly and easy to understand. The authors concluded that there would be further study to determine the functionality of microorganisms they recovered from the gut.
Specific comments:
Below are a few, minor comments and/or suggestions to be addressed:
Introduction (page 2)
Line 79
… to understand phisiological … à should be physiological
Materials and Methods (page 3)
Lines 82-89
How many samples were taken per species? Any randomization being done when selecting the samples?
Line 91
Figure 1. Add fonts in bold à Regions across Mexico where Telenomus …
Line 95
Sixty female adults were dissected … Was it 60 per species or 60 total? Please clarify.
Results (page 4-5)
Line 139
Figure 2. à switch the word order to: … phylum (A), and at class (B) levels …
Line 142
Figure 3. à switch the word order to: …order (A), and family (B) levels …
Author Response
Please see the attachment,
